# Investigation of the Relationship between Cardiovascular Biomarkers and Brachial–Ankle Pulse Wave Velocity in Hemodialysis Patients

**DOI:** 10.3390/jpm12040636

**Published:** 2022-04-15

**Authors:** Ping-Ruey Chou, Pei-Yu Wu, Ping-Hsun Wu, Teng-Hui Huang, Jiun-Chi Huang, Szu-Chia Chen, Su-Chu Lee, Mei-Chuan Kuo, Yi-Wen Chiu, Ya-Ling Hsu, Jer-Ming Chang, Shang-Jyh Hwang

**Affiliations:** 1School of Medicine, College of Medicine, Kaohsiung Medical University, Kaohsiung 80708, Taiwan; u105025047@gap.kmu.edu.tw; 2Department of Internal Medicine, Kaohsiung Municipal Hsiao-Kang Hospital, Kaohsiung Medical University, Kaohsiung 81267, Taiwan; wpuw17@gmail.com (P.-Y.W.); karajan77@gmail.com (J.-C.H.); scarchenone@yahoo.com.tw (S.-C.C.); 3Division of Nephrology, Department of Internal Medicine, Kaohsiung Medical University Hospital, Kaohsiung 80708, Taiwan; sfestg3329@gmail.com (T.-H.H.); suchle5910@gmail.com (S.-C.L.); mechku@kmu.edu.tw (M.-C.K.); chiuyiwen@kmu.edu.tw (Y.-W.C.); jemich@kmu.edu.tw (J.-M.C.); sjhwang@kmu.edu.tw (S.-J.H.); 4Faculty of Medicine, College of Medicine, Kaohsiung Medical University, Kaohsiung 80708, Taiwan; 5Graduate Institute of Medicine, College of Medicine, Kaohsiung Medical University, Kaohsiung 80708, Taiwan; yainghsu@kmu.edu.tw

**Keywords:** brachial–ankle pulse wave velocity (baPWV), cardiovascular (CV) biomarkers, hemodialysis

## Abstract

Brachial–ankle pulse wave velocity (baPWV) and cardiovascular (CV) biomarkers are correlated with clinical cardiovascular diseases (CVDs) in patients with kidney disease. However, limited studies evaluated the relationship between baPWV and CV biomarkers in hemodialysis patients. This study investigated the relationship between circulating CV biomarkers and baPWV in patients on hemodialysis. Hemodialysis patients were enrolled between August 2016 and January 2017 for the measurement of baPWV, traditional CV biomarkers, including high-sensitivity troponin-T (hsTnT) and N-terminal pro-B-type natriuretic peptide (NT-proBNP), and novel CV biomarkers, including Galectin-3, Cathepsin D, placental growth factor, Endocan-1, and Fetuin-A. The independent association was assessed by multivariate-adjusted linear regression analysis to control for potential confounders. The final analysis included 176 patients (95 men and 81 women) with a mean age of 60 ± 11 y old. After adjusting for age and sex, hsTnT (*p* < 0.01), NT-proBNP (*p* = 0.01), Galectin-3 (*p* = 0.03), and Cathepsin D (*p* < 0.01) were significantly directly correlated with baPWV. The direct correlation with baPWV existed in multivariable linear regression models with a β of 0.1 for hsTnT and 0.1 for Cathepsin D. The direct relationship between baPWV and CV biomarkers, particularly with hsTnT and Cathepsin D, may be helpful for risk stratification of hemodialysis patients.

## 1. Introduction

Chronic kidney disease (CKD) is a prevalent global health issue, with a prevalence rate of 15%, and accounts for about 1.1 million deaths annually [1,2]. Patients with end-stage renal disease (ESRD) have higher morbidity and mortality than those with cardiovascular diseases (CVDs) [3]. Despite ongoing breakthroughs in treatment, CVDs are now recognized as the primary cause of death in 54–58% of hemodialysis patients due to major adverse cardiovascular events (MACEs), including cerebrovascular disease or coronary artery disease (CAD) [4]. Better tools to stratify chronic hemodialysis patients at higher cardiovascular risk can help clinicians improve such patients’ clinical outcomes.

Several CV biomarkers are considered to provide more information regarding progressed kidney injury and cardiovascular risk to predict MACEs in managing patients with ESRD [5]. High-sensitivity troponin-T (hsTnT) and N-terminal pro-B-type natriuretic peptide (NT-proBNP) are traditional CV biomarkers highly associated with CVD in ESRD patients. In recent years, novel CV biomarkers, such as Annexin A5, Cystatin C, Plasminogen activator inhibitor-1 (PAI-1), P-Selectin, Tissue inhibitor of metalloproteinase (TIMP-1), Endocan, placental growth factor (PLGF), Galectin-3, Fetuin-A, and Cathepsin D have been studied on the clinical prognostic prediction of CV risk in different clinical settings [6,7,8,9,10,11,12,13,14,15,16,17]. Among them, Endocan [8,9], placental growth factor (PLGF) [10,11], Galectin-3 [12,13], Fetuin-A [14,15], and Cathepsin D [16,17] have been demonstrated to act as important CV risk predictors in the CKD patients; therefore, they were selected to be analyzed in our study. The concept of adaptive enrichment design leads to the future use of a biomarker in decisions regarding appropriate treatment [18].

Brachial–ankle pulse wave velocity (baPWV) is an easily applied and noninvasive measurement using an independent cuff on four limbs based on an oscillometric approach. Increased baPWV correlates with a high tendency of arterial stiffness [19], one of the underlying mechanisms of MACEs in ESRD [20,21,22]. A high baPWV is related to CV risk in pre-dialysis CKD patients [23], but its role as an indicator of CVD progression in ESRD has not been established [24].

CV biomarkers and baPWV are well-validated methods that can be easily measured in the hemodialysis unit. However, the significance of the relationship between CV biomarkers and baPWV in ESRD patients has not been well-clarified, and limited studies are focusing on the clinical efficacy of the combined use of baPWV with other biomarkers in the ESRD setting. Thus, the present study aimed to analyze the association between circulating CV biomarkers and baPWV in hemodialysis patients to investigate the underlying importance of CV risk prediction in ESRD patients.

## 2. Materials and Methods

### 2.1. Subjects

The study was conducted in the dialysis clinic of a provincial hospital in southern Taiwan between August 2016 and January 2017. In total, 193 patients who underwent hemodialysis 3 times a week using high-flux dialyzers were enrolled. Each hemodialysis session lasted 3.5–4 h, with a 500 mL/min dialysate flow rate and 250–300 mL/min blood flow rate. The Institutional Review Board of Kaohsiung Medical University Hospital approved the research protocol (KMUHIRB-E(I)-20160095), and all patients provided written informed consent. All clinical investigations were according to the Declaration of Helsinki’s standards and principles.

### 2.2. Comorbidity, Clinical Data, and Traditional CV Biomarkers

All patients’ sociodemographic data (age, gender, and smoking), dialysis vintage, dialysis access (fistula vs. graft), medical history, and biochemical data were obtained from the electronic health care system records. BMI was calculated by dividing patient weight by height squared. A blood pressure of 140/90 mmHg or greater, or the usage of antihypertensive drugs, was considered hypertension, with diabetes defined as an HbA1C of 6.5% or greater or the use of antidiabetics. Physicians’ diagnoses were used to determine a patient’s history of hyperlipidemia, CAD, and cerebrovascular illness. Hemoglobin, albumin, low-density lipoprotein cholesterol (LDL-C), ion calcium, phosphate, C-reactive protein, and single pool Kt/V (dialysis clearance) were recorded. The chemiluminescence technique (Roche Diagnostics Instrument) was used to evaluate two classic CV biomarkers, hsTnT and NT-proBNP (Appendix A).

### 2.3. Measurement of BaPWV

BaPWV was measured using an automated waveform analyzer (VP-1000, Colin, Komaki, Japan) 10–30 min before the hemodialysis session [25]. Transmission distance/transmission time was used to compute the baPWV, with the highest bilateral baPWV values used to determine the representative value for this analysis.

### 2.4. Multiplex Analysis of Novel CV Biomarkers

Serum levels of Endocan-1, PLGF, Fetuin-A, Cathepsin D, and Galectin-3 were evaluated using the MILLIPLEX MAP technology (Luminex) (Appendix A). Blood samples were taken using the arteriovenous shunt before the planned hemodialysis session in the middle of the week after an overnight fast. The serum was collected by centrifugation at 1500 rpm for 10 min at 4 °C and stored at −80 °C until use. The Luminex (Millipore, St. Charles, MO, USA), which combines the concept of a sandwich immunoassay with fluorescent-bead-based technology, allows for individual and multiplex analysis of several analytes in a single microtiter well. The multiplex immunobead tests were performed in a 96-well microplate format according to the manufacturer’s instructions. Briefly, PBS/bovine serum albumin was used to block the filter-bottom, 96-well microplate (Millipore, Billerica, MA, USA) for 10 min; then, 5-fold serial dilutions were prepared from the premixed standards to generate a 7-point standard curve. In duplicate wells, 50 µL of standards or patient sera were mixed with 50 µL of the bead mixture and incubated for one hour at room temperature. The wells were washed three times with wash buffer using a vacuum manifold and then incubated in the dark for 45 min with steady shaking with phycoerythrin (PE)-conjugated secondary antibody. The wells were washed twice, and assay buffer was added to each well (Millipore, St. Charles, MO, USA). The median fluorescence intensity was measured, and the sample concentrations were determined by the five-parameter logistic curve fitting approach through the Milliplex Analyst Software (Viagene Tech, Carlisle, MA, USA).

### 2.5. Statistical Analysis

Descriptive data are presented as mean and standard deviation (SD) or percentages as appropriate. The Kolmogorov–Smirnov test was used to determine if the data were normally distributed. The chi-square test for categorical variables, the independent *t*-test for continuous variables with nearly normal distribution, or the Mann–Whitney U test for continuous variables with skewed distribution was used to assess differences between groups. Multiple CV biomarkers linked to baPWV were screened and assessed using linear regression adjusted for age and gender. The distribution of CV biomarker values and baPWV were plotted, and Spearman rank correlation coefficients were calculated. A multivariate-adjusted linear regression analysis was used to investigate the connection between serum CV biomarkers and baPWV by controlling for relevant confounders to find independent variables. Stepwise procedures were used to identify relevant demographic factors, comorbidities, and clinical laboratory data to control variables, with a *p* < 0.05 for model entrance and a *p* > 0.1 for model removal. SAS statistical software was used for all statistical analyses (version 9.4; SAS Institute Inc., Cary, NC, USA), and the data are presented as a β with 95% CI. Statistical significance was defined as a two-tailed *p* < 0.05.

## 3. Results

### 3.1. Study Flowchart and Baseline Characteristics of Patients

The study involved 193 patients over the age of 30 y old who had been on dialysis for at least 90 d. After excluding patients who refused to complete the ankle–brachial index (ABI)-form system (*n* = 5), patients with atrial fibrillation (*n* = 4), patients with bilateral below-the-knee amputations (*n* = 3), and patients who had been hospitalized for four weeks before research enrolment (*n* = 5), the final analysis included 176 patients (95 men and 81 women) (Figure 1).

The patients’ baseline characteristics are presented in Table 1, showing that they had a mean age of 60 ± 11 y old and an average of 91.8 ± 64.3 mo hemodialysis vintage. The mean systolic and diastolic blood pressure was 156 ± 25.5 mmHg and 81.5 ± 14.6 mmHg, respectively. The median blood levels of clinical laboratory data were 10.4 mg/dL hemoglobin, 3.9 mg/dL albumin, 87.5 mg/dL LDL-C, 4.7 mg/dL ion calcium, 4.7 mg/dL phosphate, 0.2 mg/L in C-reactive protein, and 1.6 single pool Kt/V (dialysis dose). The median blood levels of clinical and novel CV biomarkers were 0.1 ng/mL hsTnT, 2.8 ng/mL NT-proBNP, 1.4 ng/mL Endocan-1, 0.01 pg/mL PLGF, 82.5 ng/mL Fetuin-A, 1 ng/mL Galectin-3, and 6.4 ng/mL Cathepsin D.

### 3.2. The Linear Association between CV Biomarkers and BaPWV

After adjusting for age and sex, hsTnT (*p* < 0.01), NT-proBNP (*p* = 0.01), Galectin-3 (*p* = 0.03), and Cathepsin D (*p* < 0.01) were significantly directly associated with baPWV in hemodialysis patients (Table 2). The direct correlation of four CV biomarkers (hsTnT, NT-proBNP, Galectin-3, and Cathepsin D in log-transformed value) and baPWV (log-transformed value) is shown in a scatter plot with a spline line (Figure 2). Furthermore, the mean values of CV biomarkers and multiple comparisons corresponding to the quartiles groups of the baPWV were found to demonstrate the dose–response effect (Figure 3).

### 3.3. Multivariable Linear Regression Model Analysis

Appendix A demonstrates the correlation between baPWV and clinical parameters and biochemical blood profiles in a simple linear regression model to determine the potential confounders for multivariable adjustment. The univariate and age-, sex-adjusted linear regression models showed a direct association between four CV biomarkers (hsTnT, NT-proBNP, Galectin-3, and Cathepsin D in log-transformed value) and baPWV (log-transformed value) levels (Model 1 and Model 2 in Table 3). Considering age, sex, smoking history, body mass index, systolic blood pressure, diastolic blood pressure, hemodialysis vintage, cause of end-stage kidney disease, diabetes mellitus, hypertension, albumin, low-density lipoprotein, iron calcium, and C-reactive protein are important contributing factors to arterial stiffness and several of them identified in single linear regression analysis (Appendix A), so these covariates were controlled in multivariable linear regression models. A significantly direct correlation remained between baPWV and hsTnT (β 0.1, 95% CI 0.04–0.2, *p* < 0.01) as well as Cathepsin D (β 0.1, 95% CI 0.02–0.1, *p* = 0.01) (Model 3 in Table 3). Furthermore, the direct association remained significantly between CV biomarkers (hsTnT and Cathepsin D) and baPWV values in the multivariable-adjusted linear regression model with stepwise covariate selection (Appendix A).

## 4. Discussion

The present study retrospectively investigated the association between baPWV and serum levels of classical and novel CV biomarkers in 176 patients undergoing long-term hemodialysis. There were direct linear correlations between CV biomarkers (hsTnT, NT-proBNP, Galectin-3, and Cathepsin D) and baPWV after adjusting for age and sex. Furthermore, an independent relationship was confirmed by the multivariable-adjusted linear regression model between baPWV and two biomarkers—hsTnT and Cathepsin D—indicating the link between peripheral vascular disease evaluation and proteins from cardiac origin. Thus, PWV not only represents a marker of peripheral vascular disease but is also a marker of cardiac disease in hemodialysis patients.

ESRD patients on maintenance hemodialysis have a 20-fold higher CV mortality rate than the general population, with an almost 10-fold higher incidence of MACE, such as cerebrovascular events or myocardial infarction [21]. Arterial stiffness is a major cause of atherosclerosis and vascular calcification, which are highly associated with CV morbidity risk in these patients [21,26]. Reduced vessel compliance and distensibility defining arterial stiffness cause greater pressure pulse transmission velocity along the arteries and are inversely correlated with PWV [27]. Compared with the direct measurement of carotid-femoral (cf) PWV and aortic PWV, baPWV can be more easily used by clinicians due to its simplicity and convenience [28]. A meta-analysis by Vlachopoulos et al. [29] demonstrated that every 1 m/s increase in baPWV was linked with a 12% increase in MACE occurrence and baPWV serves as a stronger predictor of prognosis in CVD and ESRD patients aged 59.04 to 63.36 y, a similar age to our study population (60 ± 11 y). However, Tripepi et al. [24] more recently showed minimal prognostic superiority of PWV over simple risk prediction scores in the hemodialysis population and questioned the accuracy of PWV alone for CV risk stratification. Similar to previous studies [30,31,32], factors such as age (*p* < 0.01), systolic blood pressure (*p* < 0.01), diastolic blood pressure (*p* < 0.01), hemodialysis vintage (*p* = 0.03), diabetes mellitus (*p* < 0.01), hypertension (*p* < 0.01), albumin (*p* < 0.01), ion calcium (*p* = 0.02), and C-reactive protein (*p* < 0.01) were significantly associated with baPWV (Appendix A), suggesting that these factors could be taken into accounts in multivariable-adjusted linear regression models (Table 3) for investigating the association between circulating CV biomarkers and baPWV.

Both hsTnT and NT-proBNP are classical CV biomarkers that play an important role in reflecting the subclinical damage and stress of the myocardium. The clinical utility of hsTnT and NT-proBNP in heart failure and CAD assessment has been well-validated, with hsTNT and NT-proBNP more frequently linked to the development of heart failure in CKD patients than CAD [5,33]. Arterial stiffness can intensify left ventricle afterload and exacerbate cardiac dysfunction over time [34], so the association between PWV and hsTnT and/or NT-proBNP has been discussed in several studies [35,36,37,38,39]. In the Atherosclerosis Risk in Communities Study, Liu et al. [37] reported a direct association between cfPWV measures and CV biomarkers in 3348 patients that were stronger for NT-proBNP than for hsTnT in the older population without previous CVD. However, for hemodialysis patients, Otsuka et al. [36] demonstrated that hsTnT, left ventricle ejection fraction < 50%, and baPWV were more powerful predictors of MACE than baPWV alone in univariate analysis. In the present study, the significantly direct association between baPWV and hsTnT remained after adjusting for confounders in the multivariable-adjusted model.

Galectin-3 is a β-galactoside-binding lectin (29–35 kDa) primarily released by activated macrophages, as well as other inflammatory cells in CV and renal systems [40,41,42]. A direct relationship between Galectin-3 and arterial stiffness, assessed by cfPWV, among hemodialysis patients was validated by Zhang et al. [12] after accounting for single pool Kt/V (an indicator of hemodialysis influence) in a multivariable regression model. Several theories have been proposed about the possible mechanisms. Galectin-3 is involved in lipid accumulation and chronic inflammation, which can be worsened in regular hemodialysis patients leading to arterial stiffness and atherosclerosis [43,44]. In our study, the direct association between baPWV and Galectin-3 was also examined in the multivariable-adjusted model, but the independent characteristics of Galectin-3 (*p* = 0.1) in predicting CV risk among hemodialysis patients were not significant.

Another novel CV biomarker is Cathepsin D, a cysteine protease enzyme released by macrophages, vascular smooth muscle cells, and endothelial cells in atherosclerotic plaques [45,46]. The secretion of Cathepsin D can be enhanced by inflammatory triggers into the extracellular matrix and can modify LDL by its hydrolytic ability to facilitate foam cell accumulation in the intima of the artery [47,48]. Cathepsin D is also involved in macrophage/foam cell apoptosis, causing plaque instability, an important determinant of MACE [49]. The pathological role of Cathepsin D in contributing to carotid intima–media thickness (CIMT) has been proposed by previous studies, and a direct correlation was reported in nondiabetic hypertensive patients [50] and hemodialysis patients [16]. Li et al. [51] indicated that both CIMT and PWV are predictors of developing atherosclerosis in hemodialysis patients, explaining the significant direct association between baPWV and Cathepsin D after multivariable-adjusted model analysis in the present study.

The pioneered findings of the present study suggest that using baPWV combined with classical CV biomarkers such as hsTnT and/or novel biomarkers such as Cathepsin D may provide more predictive information on CV morbidity and mortality in the ESRD population requiring regular hemodialysis. Moreover, baPWV, hsTnT, and Cathepsin D can also be beneficial in managing chronic hemodialysis patients as therapeutic indicators of ameliorating arterial stiffness. Large population trials are required in the future to determine if combining baPWV, hsTnT, and Cathepsin D improve CV risk prediction and reduce MACEs in ESRD patients on hemodialysis.

This study has several limitations. First, a causal association could not be determined due to the cross-sectional study design, so longitudinal investigations should be performed. Second, this study was limited to a single regional hospital, restricting the number of patients and their selection. Furthermore, because peritoneal dialysis patients were not included in this study, the findings may not be relevant to peritoneal dialysis patients. Third, the baPWV and bone turnover biomarkers were only measured once, which might have resulted in misclassification. Fourth, not all potential variables, such as food habits, genetic factors, medicines, and whole blood viscosity, were included in this study. For whole blood viscosity, in particular, Kobayashi et al. [52] applied the transit time of heparinized whole blood through slit pores as a whole blood viscosity indicator with baPWV to suggest the role of blood rheology in the progression of atherosclerosis in hemodialysis patients. Years later, Jung et al. [53] validated the impact of changes in systolic and diastolic whole blood viscosity on the overall survival of ESRD patients during hemodialysis. We will refer to published measurements to investigate the association between blood viscosity and CV biomarkers in hemodialysis patients in our future studies.

## 5. Conclusions

This study demonstrated a significant direct association between baPWV and CV biomarkers, especially hsTnT and Cathepsin D, in hemodialysis patients. The clinical use of baPWV combined with these CV biomarkers may serve as effective CV risk predictors or therapeutic indicators for the ESRD population undergoing regular hemodialysis.

## Figures and Tables

**Figure 1 jpm-12-00636-f001:**
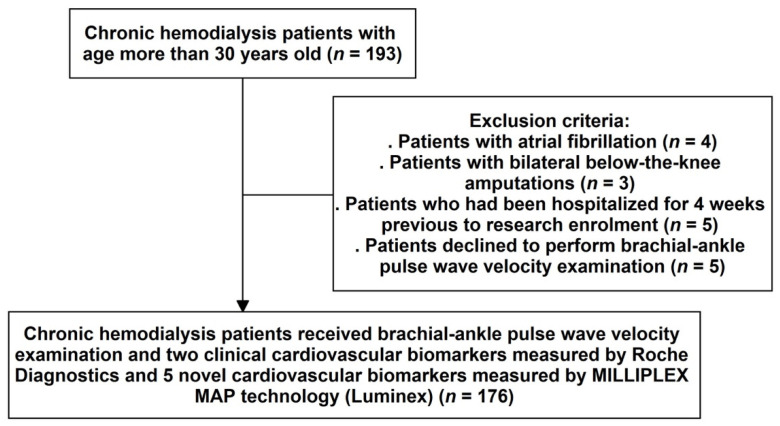
Sample derivation flowchart.

**Figure 2 jpm-12-00636-f002:**
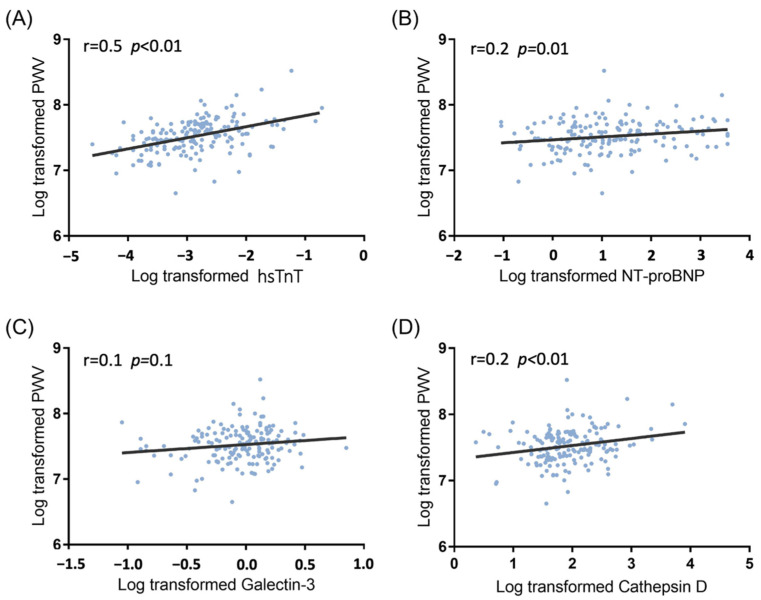
Scatter plots with spline lines demonstrated selected CV biomarkers’ direct association with brachial–ankle pulse wave velocity in hemodialysis patients: (**A**) high-sensitivity troponin T (r = 0.5, *p* < 0.01); (**B**) N-terminal pro-brain natriuretic peptide (r = 0.2, *p* = 0.01); (**C**) Galectin 3 (r = 0.1, *p* = 0.1); (**D**) Cathepsin D (r = 0.2, *p* < 0.01).

**Figure 3 jpm-12-00636-f003:**
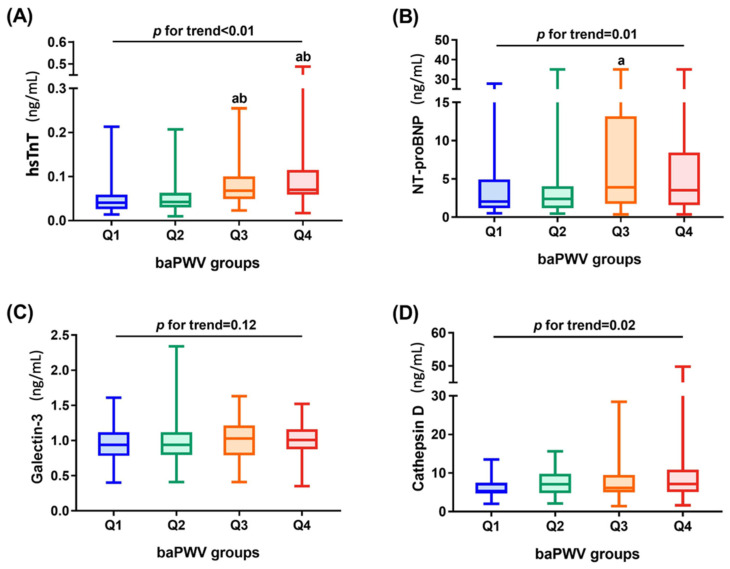
The mean values of CV biomarkers in quartile groups of brachial–ankle pulse wave velocity among hemodialysis patients: (**A**) high-sensitivity troponin T (*p* for trend < 0.01); (**B**) N-terminal pro-brain natriuretic peptide (*p* for trend 0.01); (**C**) Galectin 3 (*p* for trend = 0.12); (**D**) Cathepsin D (*p* for trend = 0.02). Multiple comparisons were performed by Dunn’s multiple comparison test (a. significant difference from Q1; b. significant difference from Q2). Statistical significance was defined as *p* < 0.05.

**Table 1 jpm-12-00636-t001:** The baseline characteristics in hemodialysis participants.

Characteristics	All Patients (*n* = 176) ^1^
Age (y old)	60 ± 11
Men, *n* (%)	95 (54%)
Smoking history, *n* (%)	64 (36.4%)
Body mass index (Kg/m^2^)	24.1 ± 3.8
Systolic blood pressure (mmHg)	156 ± 25.5
Diastolic blood pressure (mmHg)	81.5 ± 14.6
Hemodialysis vintage (mo)	91.8 ± 64.3
Arteriovenous shunt	
Arteriovenous fistula, *n* (%)	156 (88.6%)
Arteriovenous graft, *n* (%)	20 (11.4%)
Cause of end-stage kidney disease	
Hypertension, *n* (%)	13 (7.4%)
Diabetes mellitus, *n* (%)	75 (42.6%)
Glomerulonephritis, *n* (%)	69 (39.2%)
Others, *n* (%) *	19 (10.8%)
Comorbidities	
Diabetes mellitus, *n* (%)	84 (47.7%)
Hypertension, *n* (%)	124 (70.5%)
Hyperlipidemia, *n* (%)	77 (43.8%)
Coronary artery disease, *n* (%)	25 (14.2%)
Cerebrovascular disease, *n* (%)	12 (6.8%)
Clinical laboratory data	
Hemoglobin (mg/dL)	10.4 (9.7, 11.1)
Albumin (mg/dL)	3.9 (3.7, 4.1)
Low-density lipoprotein cholesterol (mg/dL)	87.5 (63.5, 107)
Ion Calcium (mg/dL)	4.7 (4.4, 5)
Phosphate (mg/dL)	4.7 (4.1, 5.3)
C-reactive protein (mg/L)	0.2 (0.1, 0.5)
Dialysis dose, Single pool Kt/V	1.6 (1.4, 1.7)
Clinical cardiovascular biomarkers	
High sensitivity troponin T (ng/mL)	0.1 (0.04, 0.09)
N-terminal pro-brain natriuretic peptide (ng/mL)	2.8 (1.5, 5.8)
Novel cardiovascular biomarkers	
Endocan-1 (ng/mL)	1.4 (1.1, 1.8)
Placental Growth Factor (pg/mL)	0.01 (0.0001, 0.03)
Fetuin-A (ng/mL)	82.5 (61, 98.5)
Galectin-3 (ng/mL)	1 (0.8, 1.2)
Cathepsin D (ng/mL)	6.4 (4.9, 9.6)

Abbreviations: Kg, kilogram; mmHg, millimeter(s) of mercury; mg, milligram; dL, deciliter; L, liter; Kt/V, dialyzer clearance of urea × dialysis time/volume of distribution of urea; ng, nanogram; mL, milliliter; pg, picogram. ^1^ Mean ± standard deviation or median (interquartile range). * Other causes of end-stage renal disease include polycystic kidney disease, tumor, systemic lupus erythematosus, gout, and interstitial nephritis.

**Table 2 jpm-12-00636-t002:** Relationship between cardiovascular disease biomarkers and brachial–ankle pulse wave velocity in hemodialysis patients.

Cardiovascular Markers	β (95%CI)	*p*-Value
hsTnT	0.2 (0.1, 0.2)	<0.01
NT-proBNP	0.04 (0.01, 0.1)	0.01
Endocan-1	0.03 (−0.1, 0.11)	0.5
PLGF	0.02 (−0.02, 0.1)	0.3
Fetuin-A	−0.03 (−0.1, 0.1)	0.6
Galectin-3	0.1 (0.01, 0.3)	0.03
Cathepsin D	0.1 (0.1, 0.2)	<0.01

The linear model is adjusted for age and sex. Abbreviations: hsTnT, high-sensitivity troponin T; NT-proBNP, N-terminal pro-brain natriuretic peptide; PLGF, placental growth factor.

**Table 3 jpm-12-00636-t003:** Association between circulating CV biomarker levels (log-transformed) and brachial–ankle pulse wave velocity (log-transformed) in hemodialysis participants using the multivariable-adjusted linear regression model.

Biomarkers	Multivariable-Adjusted Linear Regression Models *					
	Model 1	*p*-Value	Model 2	*p*-Value	Model 3	*p*-Value
hsTnT	0.2 (0.1, 0.2)	<0.01	0.2 (0.1, 0.2)	<0.01	0.1 (0.04, 0.2)	<0.01
NT-proBNP	0.04 (0.01, 0.1)	0.01	0.04 (0.01, 0.1)	0.01	0.001 (−0.03, 0.03)	0.4
Galectin-3	0.1 (−0.01, 0.3)	0.1	0.1 (0.01, 0.3)	0.03	0.1 (−0.04, 0.2)	0.1
Cathepsin D	0.1 (0.03, 0.2)	0.004	0.1 (0.1, 0.2)	<0.01	0.1 (0.02, 0.1)	0.01

Abbreviations: hsTnT, high-sensitivity troponin T; NT-proBNP, N-terminal pro-brain natriuretic peptide; * Multivariable linear model demonstrated as a β with 95% CIs. Model 1 is an unadjusted model without confounders adjustment. Model 2 is adjusted for age and sex. Model 3 is adjusted for age, sex, smoking history, body mass index, systolic blood pressure, diastolic blood pressure, hemodialysis vintage, cause of end-stage kidney disease, diabetes mellitus, hypertension, albumin, low-density lipoprotein, iron calcium, and C-reactive protein.

## Data Availability

The data presented in this study are available on request from the corresponding authors. The data are not publicly available due to privacy issues.

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
