# Peer review of "Investigation of the Relationship between Cardiovascular Biomarkers and Brachial–Ankle Pulse Wave Velocity in Hemodialysis Patients"

_jpm, 2022, doi:10.3390/jpm12040636_

Round 1
Reviewer 1 Report
Proposed paper is interesting and well written. However, some revisions are needed before it can be accepted for pubblication:
- The main determinants of arterial stiffness are age and BP (as an example see Blood Press. 2018 Feb;27(1):32-40. or Hypertension. 2013; 62(5): 934–941.) so also correction for SBP should be inserted in the model. Furthermore please insert into the model also all the factor that could influence both baPWV and biomarkers such as smoking, BMI, DM and LDL. If results doesn't remain significant please modify accordingly the paper. The other eventual variables can be selected by stepwise but the aforementioned should be forced into the models.
- Why this biomarkers have been chosen? in fact also annexin A5 (J of Hypertension. 2017 Jan;35(1):154-161. ) or cystatin C, PAI-1, P-Selectin, TIMP-1 (Blood Press. 2018 Oct;27(5):262-270.) are important CV biomarkers. Please clarify the selection of the biomarkers in the introduction.
- The plot you use in figure 2 is more frequently used for HR/OR/RR than for linear regression model (both CV biomarkers and baPWV are continuous variables). Please leave important data (beta, 95%CI and p-values) in a table instead than in a figure.
Author Response
Reviewer 1 Report:
- The main determinants of arterial stiffness are age and BP (as an example see Blood Press. 2018 Feb;27(1):32-40. or Hypertension. 2013; 62(5): 934–941.) so also correction for SBP should be inserted in the model. Furthermore please insert into the model also all the factor that could influence both baPWV and biomarkers such as smoking, BMI, DM and LDL. If results doesn't remain significant please modify accordingly the paper. The other eventual variables can be selected by stepwise but the aforementioned should be forced into the models.
Response: Thank you for your suggestion. We have added Model 3 in Table 3 to take variables like age, sex, systolic blood pressure, smoking, body mass index, diabetes mellitus, and low-density lipoprotein into account, the relevant description and explanation are in the Result section (line 6-10 in 3.3. part).
- Why this biomarkers have been chosen? in fact also annexin A5 (J of Hypertension. 2017 Jan;35(1):154-161. ) or cystatin C, PAI-1, P-Selectin, TIMP-1 (Blood Press. 2018 Oct;27(5):262-270.) are important CV biomarkers. Please clarify the selection of the biomarkers in the introduction.
Response: Thank you for your reminder. We have added an extra explanation in the Introduction section (line 5-12 in paragraph 2) to enhance the rationale for our selection of novel CV biomarkers.
- The plot you use in figure 2 is more frequently used for HR/OR/RR than for linear regression model (both CV biomarkers and baPWV are continuous variables). Please leave important data (beta, 95%CI and p-values) in a table instead than in a figure.
Response: Thank you for your advice. We have used Table 2 to present these important data accordingly.
Reviewer 2 Report
Chou et al., investigated the relationship between cardiovascular biomarkers and brachial-ankle pulse wave velocity in hemodialysis patients. The study design is poor and lack novelty. The only significance could be the clinical study. Authors needs to evaluate more related parameters and discuss accordingly.
What about biochemical blood profile and its correlation with hemodialytic patients? discuss more with reference to recent literature accordingly.
Blood viscosity and related parameters?
Arteriosclerotic parameters?
Please expand introduction and discussion part and explain why is this study is important, and could be of interest to the readers; as a lot of similar studies are already available.
Author Response
Reviewer 2 Report:
- Chou et al., investigated the relationship between cardiovascular biomarkers and brachial-ankle pulse wave velocity in hemodialysis patients. The study design is poor and lack novelty. The only significance could be the clinical study. Authors needs to evaluate more related parameters and discuss accordingly.
Response: Thank you for your feedback. In this study, we aimed to provide more information for clinicians about the potential role of several CV biomarkers (hsTNT, NT-proBNP, Galectin-3, Cathepsin D, PLGF, Endocan-1, and Fetuin-A) with baPWV being CV risk predictor and treatment indicators in hemodialysis patients. We will seriously value your comments to analyze other parameters with baPWV in CKD settings in our future studies.
- What about biochemical blood profile and its correlation with hemodialytic patients? discuss more with reference to recent literature accordingly.
Response: Thank you for your concern. The biochemical blood profile in Table 1 (Hb, Albumin, LDL, Ion calcium, Phosphate, and CRP) was the routine record in our hemodialysis care. Clinicians can leverage these data to adjust the treatment as needed. We have also summarized the correlation between baPWV and CV risk factors other than CV biomarkers including biochemical profiles in Supplementary Table 2 to determine the correlation between clinical factors and baPWV. Relevant descriptions and discussion were in the Result section (line 1-3 in 3.3 part) and the Discussion section (line 15-20 in paragraph 2).
- Blood viscosity and related parameters?
Response: Thank you for your reminder. We have discussed the importance of blood viscosity with key references in the Discussion section (line 7-15 in paragraph 7).
- Arteriosclerotic parameters?
Response: Thank you for your reminder. Our another cohort study is ongoing that applies several vascular calcification scores to assess atherosclerosis in ESRD. We are working on and looking forward to publishing our results as soon as possible.
- Please expand introduction and discussion part and explain why is this study is important, and could be of interest to the readers; as a lot of similar studies are already available.
Response: Thank you for your suggestion. We have rephrased the justification/rationale of our study in the Introduction section (line 2-5 of paragraph 4) and the Discussion section (line 1-6 in paragraph 6).
Reviewer 3 Report
The authors investigated the relationship between circulating cardiovascular biomarkers and brachial-ankle pulse wave velocity (baPWV) in patients on hemodialysis.
1) The major issue with this paper is that the entire paper looks like a statistical/mathematical paper. Where is the CV biomarker data? I don't think readers are interested in the betas, thetas, kappas, etc.
What is important is to show how the given biomarkers of CV change as the baPWV changes. Perhaps, the authors can divide the baPWV into quartiles or some meaningful clinical cut-off values and then generate the adjusted means of various CV biomarkers. This way readers can see the change in CV biomarker when the baPWV change from one category to another.
2) Authors wrote 'positive association'. I think this terminology is misleading. It should be stated/changed to "direct association/relation" or inverse association for the negative association.
3) Several references were ignored. The introduction needed to be enhanced with updated references.
4) Justification/rationale is not strong.
5) Organization: The text of the paper, the title of tables/figures are written in a seamless fashion. This leads to confusion as to where the text ends and where the title begins.
6) Figure 1: Last 2 boxes are not needed. Please remove those 2. There were 4 exclusion criteria were used. These 4 should be removed in a step-wise fashion in the chart. Now all 4 are clumped into one category.
7) Authors stated that they used "multivariate regression" but they never explain what these multi variables are.
8) All tables and figures should be shunted to the end of the paper. Each table and figure should be on a new page.
9) Methods of various biomarkers are very poorly described.
Minor
10) Figure caption (title) should be given at the bottom of the figure, right underneath it. However, the title for the table should be given on the top of the table.
11) Percent should be changed to '%'.
12) For the data with 1 to 2 digits, use 1 decimal. For 3 or more digits, no need to use decimals at all. If the data are less than one, one can use 2 decimals.
13) Tables are stand-alone and self-sustaining. That means a reader should understand the data presented in the table without referring to the text of the manuscript. So, please add more footnotes at the bottom of the table with appropriate superscripts embedded in the text of the table. Also, the table footnote should contain a type of statistical test used, abbreviations used, whether the data were mean ± SD or SE, and the significance level.
14) Like Tables, figures are also stand alone. Please give a brief title (legend) at the bottom of the figure. Also, describe the data presented in the figure in a few sentences next to the title of the figure as one paragraph. This should also include data presented (mean±SD or mean±SE), sample size, statistical tests used, or significance level.
Author Response
Reviewer 3 Report:
1) The major issue with this paper is that the entire paper looks like a statistical/mathematical paper. Where is the CV biomarker data? I don't think readers are interested in the betas, thetas, kappas, etc.
What is important is to show how the given biomarkers of CV change as the baPWV changes. Perhaps, the authors can divide the baPWV into quartiles or some meaningful clinical cut-off values and then generate the adjusted means of various CV biomarkers. This way readers can see the change in CV biomarker when the baPWV change from one category to another.
Response: Thank you for your comment. We have converted our data accordingly in Figure 3, and the relevant description was in the Result section (line 5-8 in 3.2. part).
2) Authors wrote 'positive association'. I think this terminology is misleading. It should be stated/changed to "direct association/relation" or inverse association for the negative association.
Response: Thank you for your suggestion. We have revised the terminology accordingly in the Abstract section (line 12-14), the Results section (line 2-3 in 3.2. part and line 4-12 in 3.3. part), the Discussion section (line 3 in paragraph 1, line 8 and 13 in paragraph 3, line 3 and 9 in paragraph 4, and line 8 and 11 in paragraph 5), and the Conclusion section (line 1).
3) Several references were ignored. The introduction needed to be enhanced with updated references.
Response: Thank you for your reminder. We have carefully checked all references and added updated references in the Introduction section.
4) Justification/rationale is not strong.
Response: Thank you for your comment. We have restated our justification/rationale in the Introduction section (line 2-5 of paragraph 4).
5) Organization: The text of the paper, the title of tables/figures are written in a seamless fashion. This leads to confusion as to where the text ends and where the title begins.
Response: Thank you for your feedback. We have applied the template of the Journal of Personalized Medicine to optimize the typesettings.
6) Figure 1: Last 2 boxes are not needed. Please remove those 2. There were 4 exclusion criteria were used. These 4 should be removed in a step-wise fashion in the chart. Now all 4 are clumped into one category.
Response: Thank you for your suggestion. We have modified the study flowchart (Figure 1) accordingly.
7) Authors stated that they used "multivariate regression" but they never explain what these multi variables are.
Response: Thank you for your comment. We have indicated selected covariates in the footer of Table 2.
8) All tables and figures should be shunted to the end of the paper. Each table and figure should be on a new page.
Response: Thank you for your suggestion. We have modified the positions of all tables and figures accordingly.
9) Methods of various biomarkers are very poorly described.
Response: Thank you for your reminder. Methods of various CV biomarkers were described in the Abstract section (line 6-9), the Materials and Methods section (line 9-11 in 2.2 part and the whole 2.4. part).
Minor
10) Figure caption (title) should be given at the bottom of the figure, right underneath it. However, the title for the table should be given on the top of the table.
Response: Thank you for your suggestion. We have revised the position of all figure captions and table titles accordingly.
11) Percent should be changed to '%'.
Response: Thank you for your feedback. We have changed “percent” into “%” in the Introduction section (line 2 in paragraph 1).
12) For the data with 1 to 2 digits, use 1 decimal. For 3 or more digits, no need to use decimals at all. If the data are less than one, one can use 2 decimals.
Response: Thank you for your suggestion. We have unified the usage of the decimal point to present our data accordingly.
13) Tables are stand-alone and self-sustaining. That means a reader should understand the data presented in the table without referring to the text of the manuscript. So, please add more footnotes at the bottom of the table with appropriate superscripts embedded in the text of the table. Also, the table footnote should contain a type of statistical test used, abbreviations used, whether the data were mean ± SD or SE, and the significance level.
Response: Thank you for your suggestion. We have added more descriptions about type of statistical test used, abbreviations used, whether the data were mean ± SD or SE, and the significance level in footnotes of Table 1 and 2.
14) Like Tables, figures are also stand alone. Please give a brief title (legend) at the bottom of the figure. Also, describe the data presented in the figure in a few sentences next to the title of the figure as one paragraph. This should also include data presented (mean±SD or mean±SE), sample size, statistical tests used, or significance level.
Response: Thank you for your reminder. We have added more descriptions of the data presented in the legends of Figures 2 and 3.
Reviewer 4 Report
This is an interesting study, but there are some questions to be elucidated.
Authors documented about comorbidity, but there is no information about primary disease of ESRD. I understood that half of patients showed DM in their cohort, but it does not mean primary disease is DM. I assumed some parts of patients were IgA nephropathy and other forms of nephropathy. Heterogeneity of their cohort should be documented.
Authors showed correlation between log-transformed baPWV and 4 markers in Figure 2 and 3. I wonder how it looks like if pearson’s or spearman’s correlation coefficient is performed, which authors documented in methods
Authors performed multivariable linear regression model analysis. I could not understand what they mean for model 1, 2 and 3.
Did model 1 include age and 4 markers as confounders?
Did model 2 include sex and 4 markers as confounders?
Did model 3 include age, sex and 4 markers and perform stepwise selection?
Please describe what kinds of confounders were included in each model.
Please show what kinds of clinical parameters were extracted as independent risk factors of baPWV. I assume that DM would be risk factor and recommend authors to perform include other parameters including DM history.
Author Response
Reviewer 4 Report:
- Authors documented about comorbidity, but there is no information about primary disease of ESRD. I understood that half of patients showed DM in their cohort, but it does not mean primary disease is DM. I assumed some parts of patients were IgA nephropathy and other forms of nephropathy. Heterogeneity of their cohort should be documented.
Response: Thank you for your suggestion. We have added the information on the causes of ESRD in Table 1.
- Authors showed correlation between log-transformed baPWV and 4 markers in Figure 2 and 3. I wonder how it looks like if pearson’s or spearman’s correlation coefficient is performed, which authors documented in methods
Response: Thank you for your comment. We have modified our scatter plots with spline lines and added the Spearman rank correlation coefficients in Figure 2.
- Authors performed multivariable linear regression model analysis. I could not understand what they mean for model 1, 2 and 3.
Did model 1 include age and 4 markers as confounders?
Did model 2 include sex and 4 markers as confounders?
Did model 3 include age, sex and 4 markers and perform stepwise selection?
Please describe what kinds of confounders were included in each model.
Response: Thank you for your feedback. We have further described all confounders in the footnote of Table 3.
- Model 1 is a crude model without confounders adjustment.
- Model 2 is adjusted for age and sex.
- Model 3 is adjusted for age, sex, smoking history, body mass index, systolic blood pressure, diastolic blood pressure, hemodialysis vintage, cause of end-stage kidney disease, diabetes mellitus, hypertension, albumin, low-density lipoprotein, iron calcium, and C-reactive protein.
- Model 4 is adjusted for stepwise procedure selected covariates in the final model as follows: hsTNT covariates selection is age, diastolic blood pressure, and blood level of ion Calcium; NT-proBNP covariates selection is age, diastolic blood pressure, diabetes mellitus, and blood level of low-density lipoprotein and C-reactive protein; Galectin 3 covariates selection is age, body mass index, systolic blood pressure, diabetes mellitus, and blood level of ion calcium and C-reactive protein; Cathepsin D covariates selection is age, body mass index, systolic blood pressure, diabetes mellitus, and blood level of ion calcium and C-reactive protein.
- Please show what kinds of clinical parameters were extracted as independent risk factors of baPWV. I assume that DM would be risk factor and recommend authors to perform include other parameters including DM history.
Response: Thank you for your recommendation. We have described all relevant clinical confounders as independent variables of Model 1-4 in the footnotes of Table 3 and taken DM history into account in Model 3.
Round 2
Reviewer 1 Report
Authors replies to all the query raised and paper can now be accepted for pubblication.
Author Response
Thanks for your reviews and comments to improve our study quality.
Reviewer 2 Report
Authors have improvised the manuscript.
Author Response

(The authors gave the same response as above.)

Reviewer 3 Report
I appreciate the changes/revisions made by the author. However, there is one major and are several minor revisions that should be made before the paper can be accepted.
Major revision:
Figure 1 is great. Please do the multiple comparison test or at least compare Q1 with Q2, Q3, and Q4 in all 4 biomarkers.
Minor revisions:
Authors need to pay attention to details. They have ignored several fundamentals in scientific writing. Authors need to go over the entire paper carefully and fix those fundamental errors. Some, I have highlighted below.
1) Please change the hs-TNT to hs-TnT throughout the paper. TNT acronym is generally used for an explosive substance.
2) Abstract: Remove the manufacturer names such as "Roche Diagnostics and MILLI- 29 PLEX® map Kits" These are not commonly mentioned in the abstract. These should be mentioned in the 'methods' section of the main paper.
3) Abstract: Please remove "substantially". I am not sure what this means. These are qualitative descriptors and these terms should be backed by data to be scientifically valid. In the absence of quantitative data, substitute with "significantly" and give a p-value at the end of this sentence to validate this statement. Modify this to:
"After adjusting for age and sex,............................ significantly, directly associated with baPWV (P ?).
"Association/relation is measured with correlation or regression". Please change to this expression throughout the paper.
3) Change "59.8 ± 11.0 years" to "60±11 y". Please simplify the data. Remove these '0' in other places too. 11.0 is the same as 11. Why extra '0'?
4) Use y for year/s, d for day/s, mo for month/s, min for minute/s etc. These are standard abbreviations commonly used.
5) Authors have used "β coefficient" throughout the paper. In fact, 'β' means regression coefficient. So just use 'β' and remove 'coefficient' throughout the paper.
6) Line 45: Abbreviate "cardiovascular (CV) diseases" as "cardiovascular diseases (CVD)" and starting the second mention, you can use the 'CVD' acronym and not "CV disease/s". CV acronym is also commonly used for "coefficient of variation".
7) Line 139: Abbreviate "95% confidence intervals (Cis)" as "95% CI".
8) Change "p-value 0.05" to "p=0.05". Remove "value".
9) Is it 'LDL" or "LDL cholesterol"?
10) Change "Figure 1. Study flowchart" to "Figure 1. Sample derivation flow chart".
11) For the data with 1 to 2 digits, use 1 decimal. For 3 or more digits, no need to use decimals at all. If the data are less than one, one can use 2 decimals.
This recommendation was not followed through. There are too many unnecessary decimals.
12) Change "Male gender" to "Men". Male and gender are redundant. On the other note, change "male to men" and "female to women" throughout the paper.
13) Table 1: Column 1: This table was built in a rush and care was not taken in writing the units. Change to:
"Men, n (%)
Hypertension, n (%)
Diabetes Mellitus, n (%)
Glomerulonephritis, n (%)
Others, n (%)* "
Do this for all other similar descriptors.
14) Change "mg/dl' to "mg/dL", pg/ml to pg/mL, l to L, etc. Here letter 'l' should be "upper case" not lower case. Do this throughout the paper.
15) Tables: There are so many unnecessary 0. For example:
Change to:
"Phosphate (mg/dL) 4.7 (4.1, 5.3)
C-reactive protein (mg/L) 0.2 (0.1, 0.5)"
Please simplify.
16) Table 2 title: The title is not appropriate and grammatically incorrect. Change to:
Table 2: Relationship between cardiovascular disease biomarkers and brachial-ankle pulse wave velocity in hemodialysis patients.
The other details should be given in the footnote. Please see my previous recommendations regarding writing footnotes. Authors should review a few papers in the literature to be familiar with the format of tables.
16) Authors keep describing "crude model". It is not a "crude model". It should be changed to "unadjusted model" throughout the paper.
17) Table 3: There is no point in presenting 4 models. It is excessive. Just present 3 models (suggestion: keep model 1, model 2 or 3, and model 4). Create another column for each model, next to the right. Please present the "p for trend" for each biomarkers, under each model, in these additional columns.
18) Referencing: The reference number should be given right after the author's name. For example, it should be written as "Kobayashi et al [53]". The reference number should not be written separately somewhere else. Please do this throughout the paper.
19) Line 138: (version 9.4; SAS Institute Inc)- Please state city, state, and country in the parenthesis.
20) Line 143: Change to "age of 30 y old".
21) Acronyms: HTN and CRP are mentioned only once. If an acronym is mentioned only once there is no need to abbreviate just expand once and leave it.
22) Have your paper reviewed by a senior scientist (who has published before) for scientific editing.
23) There are too many abbreviations in the paper. Unabbreviate LV, HF
Author Response
Dear Reviewer:
Thank you very much for your comments on our manuscript. We have carefully considered the comments of the reviewers and revised our manuscript accordingly. The revised parts have been highlighted in the manuscript. Our responses to the reviewers' comments are as follows.
Reviewer 3 Report:
I appreciate the changes/revisions made by the author. However, there is one major and are several minor revisions that should be made before the paper can be accepted.
Major revision:
Figure 1 is great. Please do the multiple comparison test or at least compare Q1 with Q2, Q3, and Q4 in all 4 biomarkers.
Response: Thank you for your suggestion. We have added the results of multiple comparison in Figure 3, and the relevant descriptions were in the Figure 3 legend and the Results section (line 5-7 in 3.2. part).
Minor revisions:
Authors need to pay attention to details. They have ignored several fundamentals in scientific writing. Authors need to go over the entire paper carefully and fix those fundamental errors. Some, I have highlighted below.
1) Please change the hs-TNT to hs-TnT throughout the paper. TNT acronym is generally used for an explosive substance.
Response: Thank you for your important reminder. We have revised the acronym accordingly throughout the paper.
2) Abstract: Remove the manufacturer names such as "Roche Diagnostics and MILLI- 29 PLEX® map Kits" These are not commonly mentioned in the abstract. These should be mentioned in the 'methods' section of the main paper.
Response: Thank you for your suggestion. We have removed "Roche Diagnostics and MILLI- 29 PLEX® map Kits" from the Abstract section.
3) Abstract: Please remove "substantially". I am not sure what this means. These are qualitative descriptors and these terms should be backed by data to be scientifically valid. In the absence of quantitative data, substitute with "significantly" and give a p-value at the end of this sentence to validate this statement. Modify this to:
"After adjusting for age and sex,............................ significantly, directly associated with baPWV (P ?).
"Association/relation is measured with correlation or regression". Please change to this expression throughout the paper.
Response: Thank you for your comment. We have revised accordingly in the Abstract section (line 11-12), and the Results section (line 2 in the 3.2. part).
4) Change "59.8 ± 11.0 years" to "60±11 y". Please simplify the data. Remove these '0' in other places too. 11.0 is the same as 11. Why extra '0'?
Response: Thank you for your suggestion. We have simplified the data and removed all extra extra '0' throughout this paper.
5) Use y for year/s, d for day/s, mo for month/s, min for minute/s etc. These are standard abbreviations commonly used.
Response: Thank you for your reminder. We have revised accordingly throughout this paper.
6) Authors have used "β coefficient" throughout the paper. In fact, 'β' means regression coefficient. So just use 'β' and remove 'coefficient' throughout the paper.
Response: Thank you for your reminder. We have revised accordingly throughout this paper.
7) Line 45: Abbreviate "cardiovascular (CV) diseases" as "cardiovascular diseases (CVD)" and starting the second mention, you can use the 'CVD' acronym and not "CV disease/s". CV acronym is also commonly used for "coefficient of variation".
Response: Thank you for your comment. We have revised accordingly throughout this paper.
8) Line 139: Abbreviate "95% confidence intervals (Cis)" as "95% CI".
Response: Thank you for your comment. We have revised it accordingly.
9) Change "p-value 0.05" to "p=0.05". Remove "value".
Response: Thank you for your comment. We have revised it accordingly.
10) Is it 'LDL" or "LDL cholesterol"?
Response: Thank you for your reminder. It is LDL cholesterol, we have revised it in the Method section (2.2. part), the Results section (3.1. part), Table 1, and Supplementary Table 2.
11) Change "Figure 1. Study flowchart" to "Figure 1. Sample derivation flow chart".
Response: Thank you for your suggestion. We have revised it accordingly.
12) For the data with 1 to 2 digits, use 1 decimal. For 3 or more digits, no need to use decimals at all. If the data are less than one, one can use 2 decimals.
This recommendation was not followed through. There are too many unnecessary decimals.
Response: Thank you for your recommendation. We have revised it throughout this paper.
13) Change "Male gender" to "Men". Male and gender are redundant. On the other note, change "male to men" and "female to women" throughout the paper.
Response: Thank you for your suggestion. We have revised it accordingly.
14) Table 1: Column 1: This table was built in a rush and care was not taken in writing the units. Change to:
"Men, n (%)
Hypertension, n (%)
Diabetes Mellitus, n (%)
Glomerulonephritis, n (%)
Others, n (%)* "
Do this for all other similar descriptors.
Response: Thank you for your reminder. We have revised it accordingly.
15) Change "mg/dl' to "mg/dL", pg/ml to pg/mL, l to L, etc. Here letter 'l' should be "upper case" not lower case. Do this throughout the paper.
Response: Thank you for your reminder. We have revised it accordingly throughout the paper.
16) Tables: There are so many unnecessary 0. For example:
Change to:
"Phosphate (mg/dL) 4.7 (4.1, 5.3)
C-reactive protein (mg/L) 0.2 (0.1, 0.5)"
Please simplify.
Response: Thank you for your concern. We have removed all unnecessary 0 to simplify our presented data accordingly throughout the paper.
17) Table 2 title: The title is not appropriate and grammatically incorrect. Change to:
Table 2: Relationship between cardiovascular disease biomarkers and brachial-ankle pulse wave velocity in hemodialysis patients.
The other details should be given in the footnote. Please see my previous recommendations regarding writing footnotes. Authors should review a few papers in the literature to be familiar with the format of tables.
Response: Thank you for your advice. We have revised it accordingly.
18) Authors keep describing "crude model". It is not a "crude model". It should be changed to "unadjusted model" throughout the paper.
Response: Thank you for your advice. We have revised it accordingly in the footnote of Table 3.
19) Table 3: There is no point in presenting 4 models. It is excessive. Just present 3 models (suggestion: keep model 1, model 2 or 3, and model 4). Create another column for each model, next to the right. Please present the "p for trend" for each biomarkers, under each model, in these additional columns.
Response: Thank you for your advice. We have revised it accordingly in Table 3 and Supplementary Table 3. The relevant description was in the Discussion section (line 6-17 in 3.3. part.).
20) Referencing: The reference number should be given right after the author's name. For example, it should be written as "Kobayashi et al [53]". The reference number should not be written separately somewhere else. Please do this throughout the paper.
Response: Thank you for your suggestion. We have revised it accordingly throughout this paper.
21) Line 138: (version 9.4; SAS Institute Inc)- Please state city, state, and country in the parenthesis.
Response: Thank you for your reminder. We have revised it accordingly.
22) Line 143: Change to "age of 30 y old".
Response: Thank you for your reminder. We have revised it accordingly.
23) Acronyms: HTN and CRP are mentioned only once. If an acronym is mentioned only once there is no need to abbreviate just expand once and leave it.
Response: Thank you for your comment. We have revised it accordingly.
24) Have your paper reviewed by a senior scientist (who has published before) for scientific editing.
Response: Thank you for your concern. We have invited several senior scientists to check our scientific editing again before submitting the revised version.
25) There are too many abbreviations in the paper. Unabbreviate LV, HF
Response: Thank you for your comment. We have revised it accordingly.
Ping-Hsun Wu, MD
on behalf of all authors
Corresponding author:
Ping-Hsun Wu, MD
Division of Nephrology, Department of Internal Medicine, Kaohsiung Medical University Hospital, Kaohsiung Medical University, 807 Kaohsiung, Taiwan
100 Shih-Chuan 1st Road Kaohsiung 807, Taiwan
Telephone number: 886-7-3121101
E-mail address: 970392@ kmuh.org.tw
Reviewer 4 Report
Authors answered my questions appropriately.
Author Response

(The authors gave the same response as above.)
